# Epstein–Barr Virus Infection of Oral Squamous Cells

**DOI:** 10.3390/microorganisms8030419

**Published:** 2020-03-16

**Authors:** Chukkris Heawchaiyaphum, Hisashi Iizasa, Tipaya Ekalaksananan, Ati Burassakarn, Tohru Kiyono, Yuichi Kanehiro, Hironori Yoshiyama, Chamsai Pientong

**Affiliations:** 1Department of Microbiology, Faculty of Medicine, Khon Kaen University, Khon Kaen 40002, Thailand; jukkris.003@gmail.com (C.H.); tipeka@kku.ac.th (T.E.); aburassakarn@gmail.com (A.B.); 2Department of Microbiology, Shimane University Faculty of Medicine, Shimane 693-8501, Japan; iizasah@med.shimane-u.ac.jp (H.I.); kanehiro@med.shimane-u.ac.jp (Y.K.); 3HPV & EBV and Carcinogenesis Research Group, Khon Kaen University, Khon Kaen 40002, Thailand; 4Division of Virology, National Cancer Center Research Institute, Tokyo 104-0045, Japan; tkiyono@ncc.go.jp

**Keywords:** Epstein–Barr virus, oral squamous cell carcinoma, differentiation of SCC cells, EBV lytic replication, cancer progression

## Abstract

The Epstein–Barr virus (EBV) is a human herpesvirus associated with various cancers. The number of reports that describe infection of EBV in oral squamous carcinoma cells is increasing. However, there is no available in vitro model to study the possible role of EBV in the development of oral squamous cell carcinoma. Herein, we report establishment of a latent EBV infection of well-differentiated HSC1 cells and poorly differentiated SCC25 cells. Viral copy numbers per cell in EBV-infected HSC1 and SCC25 cells are 2 and 5, respectively. Although the EBV copy number was small, spontaneous viral replication was observed in EBV-infected HSC1 cells. Contrarily, infectious viral production was not observed in EBV-infected SCC25 cells, despite containing larger number of EBV genomes. Chemical activation of cells induced expression of viral lytic BZLF1 gene in EBV-infected HSC1 cells, but not in EBV-infected SCC25 cells. EBV infection activated proliferation and migration of HSC1 cells. However, EBV-infection activated migration but not proliferation in SCC25 cells. In conclusion, EBV can infect squamous cells and establish latent infection, but promotion of cell proliferation and of lytic EBV replication may vary depending on stages of cell differentiation. Our model can be used to study the role of EBV in the development of EBV-associated oral squamous cell carcinoma.

## 1. Introduction

The Epstein–Barr virus (EBV) is a human gamma-herpesvirus that infects mostly through the oral route and is transmitted to other people via saliva, since its infectious particles are shed in the saliva of the majority (>90%) of adults [1]. Primary infection of EBV is mild or asymptomatic in early childhood, but often manifests itself as infectious mononucleosis (IM) in adolescents. After the initial infection, EBV causes lifelong persistent infection in cells, which is called latent infection and is classified in four distinct types, from 0 to III. Latent infection is a viral strategy to increase its progenies by promoting the proliferation of the infected host cells; this sometimes drives oncogenic transformation to generate Burkitt’s lymphoma, Hodgkin’s disease, diffuse large B cell lymphomas (DLBCLs), NK/T cell lymphomas, and others [2]. EBV infects not only lymphocytes, but also epithelial cells, thus causing several types of EBV-related epithelial malignancies, including nasopharyngeal carcinoma (NPC) and EBV-associated gastric carcinomas (EBVaGC) [3,4].

Oral hairy leukoplakia is caused by EBV usually in immunocompromised persons, and appears as a white patch on the side of the tongue [1,5]. Similar EBV infection may occur on the buccal mucosa, soft palate, pharynx, or esophagus. The histological appearance is created by hyperkeratosis and epithelial hyperplasia, where active EBV replication is apparent [6]. Epidemiological studies showed that not only oral hairy leukoplakia, but also oral squamous cell carcinoma (OSCC) sometimes associate with EBV, with differences in etiological, geographic, and ethnic background [7,8,9,10]. EBV infection was detected in 82.5% of OSCC by microarray technique, focusing on EBV-specific genes such as EBV nuclear antigens (EBNAs), latent membrane protein (LMP) 2A, LMP2B, and structural proteins [11]. EBV genome and latent viral gene products were detected in normal epithelia, epithelial dysplasia, and squamous cell carcinoma (SCC) of the oral cavity [12]. Moreover, EBNA2 and LMP1 were also expressed in 50.2% and 10.7% of OSCC cases, respectively. A meta-analysis revealed that EBV association increased the risk for OSCC by 5.03 fold (95% CI; 1.80–14.01) [13].

The oncogenic property of EBV can be observed when in vitro infection of EBV immortalizes and transforms primary B lymphocytes [14]. Although there is no model of transformation in epithelial EBV infection, episomal EBV genomes are detected in every cell and maintained at high levels in early invasive NPC tumor cells [15,16]. EBV episomes within epithelial tumor cells are derived from the same ancestral virus that has established latent infection [17,18]. Moreover, the expression of viral latent gene products, such as EBV-encoded small RNAs (EBERs), BART microRNAs, and latent membrane proteins (LMP1 and LMP2A) confers advantages to infected cells, including proliferation speed, apoptotic resistance, anchorage-independence, and invasiveness [2,3,4].

Though some observed cases of EBV-positive epithelial tumor already suggested the association of EBV with oral carcinogenesis, the present study shows EBV infection in OSCC cells from patient samples. Moreover, we analyze the involvement of EBV in the pathogenesis and progression of OSCCs by creating an in vitro model of EBV infections using SCC cells.

## 2. Materials and Methods

### 2.1. Patient Specimens

One-hundred and sixty-five formalin-fixed paraffin-embedded (FFPE) OSCC samples, diagnosed at Srinakarind Hospital, were retrieved from materials archived in the Department of Pathology, Faculty of Medicine, Khon Kaen University, Thailand. In each case, histological diagnosis was made by a pathologist using FFPE tissue sections. The use of FFPE OSCC archived specimens was approved by the Khon Kaen University Ethics Committee for Human Research (HE581211, 9 June 2015).

### 2.2. Cell Culture

Akata cells infected with an EBV recombinant with enhanced green fluorescent protein (eGFP) gene and neomycin resistant gene at BXLF1 gene (EBV-eGFP) were used as producers of wild-type EBV [19]. Human tongue SCC-derived SCC25 cells and human skin SCC-derived HSC1 cells were cultured in Dulbecco’s Modified Eagle’s Medium (DMEM)/Ham’s F12 (Wako, Osaka, Japan). Gastric carcinoma cells, as in AGS (-), AGS-CD21, and EBV-eGFP positive AGS (AGS (+)), and B cells (B95-8, Raji, EBV-eGFP positive Akata (Akata EBV), and Daudi (-)) were cultured in RPMI-1640 medium (Sigma, St. Louis, MO, USA). All cells were supplemented with 10% fetal bovine serum (FBS) (HyClone^TM^, GE Healthcare, Chicago, IL, USA) and penicillin-streptomycin solution (Nacalai Tesque Inc., Kyoto, Japan). Cells were cultured at 37 °C in a 5% CO_2_ incubator.

### 2.3. EBV Detection by PCR and EBER in Situ Hybridization (ISH)

FFPE OSCC tissues samples were cut at 4 µm thickness and treated for DNA extraction by using QIAamp DNA Mini Kit (Qiagen, Hilden, Germany) according to the manufacturer’s protocol for tissue specimens. DNA quality was checked by amplification of β-actin gene (Table 1). EBV infection in FFPE OSCC tissues was determined by detecting EBNA1 and LMP1 DNA (Table 1). EBV-positive FFPE OSCC tissues were used to confirm the infection of EBV by using EBER in situ hybridization (ISH) as described [20].

### 2.4. Establishment of EBV-Positive SCC Cells

EBV-positive SCC cells were established with the cell-to-cell infection method [21]: Akata EBV cells at 2 × 10^6^ cells were treated with 0.5% anti-human IgG (Dako, Glostrup, Denmark) and cultured for 48 h in a 5% CO_2_ incubator. HSC1 and SCC25 cells were seeded at 1 × 10^5^ cell/well into 6-well plates for a 9 h incubation and washed with phosphate buffered saline (PBS) for 3 times and co-cultured for 48 h with the previously prepared Akata EBV cells. HSC1 and SCC25 cells were then washed 5 times with PBS to remove Akata EBV cells, and subsequently cultured for 48 h in a 5% CO_2_ incubator. EBV-infected cells expressing eGFP were observed under fluorescent microscope and subjected to drug selection using 300 and 200 µg/mL of G418 (Promega, Madison, WI, USA) for HSC1 and SCC25 cells, respectively. Established EBV-positive HSC1 cells and EBV-positive SCC25 cells were named HSC1 EBV cells and SCC25 EBV cells, respectively.

### 2.5. Western Blotting

Proteins were extracted from cells using radioimmunoprecipitation buffer. Twenty µg of total proteins were subjected to SDS-PAGE and transferred onto PVDF membrane (GE Healthcare, Chicago, IL, USA). Membranes were then probed with primary antibodies for EBV proteins, including EBNA1 (Santa Cruz Biotech, Santa Cruz, CA, USA), EBNA2 (Abcam, Cambridge, UK), LMP1 (Dako, Glostrup, Denmark), LMP2A (Santa Cruz Biotech, Santa Cruz, CA, USA) and β-actin (A1978; Sigma, St. Louis, MO, USA). After probing, membranes were incubated with horseradish peroxidase (HRP)-conjugated goat anti-mouse IgG or HRP-conjugated anti-sheep IgG (GE Healthcare, Chicago, IL, USA) and visualized using Image Quant LAS 4000mini (GE Healthcare, Chicago, IL, USA).

### 2.6. Quantitative Reverse Transcription PCR (qRT-PCR)

Total RNA was extracted from cells using ISOGEN reagent (Nippon Gene, Tokyo, Japan) and 1 µg of RNA was used to synthesize cDNA by SuperScript® III Reverse Transcriptase (Invitrogen, Carlsbad, CA, USA) and random hexamer (Promega, Madison, WI, USA). Gene expression was quantified by qRT-PCR assay using SsoAdvanced^TM^ SYBR® Green Supermix and CFX Connect real-time PCR System (Bio-Rad, Hercules, CA, USA) or RT-PCR assay (KAPA2G Fast PCR kit, Sigma, St. Louis, MO, USA). Glyceraldehyde 3-phosphate dehydrogenase (GAPDH) or β-actin were used as an internal control. Relative mRNA expression level was quantified using the 2^-ΔΔCT^ method (Table 1).

### 2.7. Quantification of EBV Copy Number

EBV copy number was quantified by qPCR. Genomic DNA was extracted from both EBV-positive and EBV-negative cells using GenElute™ Mammalian Genomic DNA Miniprep Kit (Sigma, St. Louis, MO, USA). Calibration curves were run in parallel and in triplicate with each analysis, using sequential dilutions of genomic DNA from the Raji cell line, because each diploid Raji cell harbors 55 EBV copies/cell. Specific primers for EBV genome were used (Table 1) [22].

### 2.8. Immunofluorescence Staining

HSC1 cells and SCC25 cells were grown in 6-well plates at 2 × 10^5^ cells/well overnight. Cells were washed with PBS and mixed with primary antibody against CD21 (BioLegend, San Diego, CA, USA) at 4 °C for 30 min. After washing, cells were stained with goat anti-mouse IgG Alexa Fluor 647 (Invitrogen, Carlsbad, CA, USA). CD21-negative AGS (-) cells and CD21-positive Daudi (-) cells were used.

AGS (+) cells, HSC1 EBV cells, and SCC25 EBV cells were grown overnight in 6-well plates at 2 × 10^5^ cells/well, then cultured for 48 h with 30 ng/mL of 12-O-tetradecanoylphorbol-13-acetate (TPA) and 3 mM sodium butyrate (SB) (Sigma, St. Louis, MO, USA) to induce lytic replication of EBV. Mouse monoclonal antibody against gp350/220 and goat anti-mouse IgG Alexa Fluor 647 (Invitrogen, Carlsbad, CA, USA) were used to stain the cells, which were then washed with 2% FBS/PBS. To eliminate dead cells, the cell suspension was stained with 7-actinomycin D (7-AAD) Viability Staining Solution (eBioscience, San Diego, CA, USA) and analyzed by flow cytometry (FACSCalibur, BD Bioscience, Franklin Lakes, NJ, USA).

AGS (+) cells, HSC1 EBV cells, and SCC25 EBV cells were fixed with cold acetone for 5 min at −20 °C and stained with primary antibodies for EBV lytic proteins. Antibodies against BMRF1 (R3), BZLF1 (BZ-1), gp110 (C30-1), and gp350/220 (C-1) were diluted 1:10 in Can Get Signal® solution A (TOYOBO, Osaka, Japan) and incubated at 37 °C for 1 h [23,24]. After primary antibody staining, cells were stained with goat anti-mouse IgG antibody conjugated with Cy3 (Thermo Fisher Scientific, Indianapolis, IN, USA) diluted with Can Get Signal® solution B. DAPI (Wako, Osaka, Japan) was used for nuclear staining. Expression and localization were examined using FV1000D laser scanning confocal microscope (Olympus, Tokyo, Japan).

### 2.9. Virus Titration by Flow Cytometry

AGS (+), HSC1 EBV, and SCC25 EBV cells were cultured in 6-well plates at 3 × 10^5^ cells/well for 24 h. After incubation, cells were treated with TPA/SB for 48 h. Culture supernatants were filtrated with 0.8 µM filter membranes and centrifuged at 16,440× *g* for 90 min. Pellets were resuspended in fresh medium to make virus suspensions. Serial dilutions of virus were added into 96-well plates containing Daudi (-) cells at 2 × 10^4^ cells/well and incubated at 37 °C, 5% CO_2_ for 48 h [24]. After incubation, cells were washed and 7-AAD was added into cell suspensions to distinguish living cells and death cells. Cell suspensions were subjected to flow cytometry to quantify the GFP-positive cells. The virus titer was obtained using the formula:

Virus titer = - In (1 - (number of eGFP positive/number of cells quantified by flow cytometry)) × number of total cells × dilution factor

### 2.10. Cell Proliferation Assay

Cell proliferation was determined with the Cell Counting Kit-8 (CCK-8, DOJINDO, Kumamoto, Japan). The suspension of HSCS1 cells, HSC1 EBV cells, SCC25 cells, and SCC25 EBV cells (100 μL/well, 5000 cells/well) was added into 96-well plates and incubated at 37 °C in a 5% CO_2_ incubator for 6, 12, 24, 48, 72 and 96 h, respectively. After incubation, cells were incubated in 10 µL/well of CCK-8 solution for 1–4 h and measured for the absorbance at 450 nm using a microplate reader (Beckman Coulter, Miami, FL, USA).

### 2.11. Wound Healing Assay

HSC1 cells, HSC1 EBV cells, SCC25 cells, and SCC25 EBV cells were seeded into 24-well plates at 2 × 10^5^ cells/well and incubated at 37 °C under 5% CO_2_ to become 90% confluent. Cells were washed 3 times with PBS. Wounds made by SPL ScarTM scratcher (SPL life sciences, Gyeonggi-do, Korea) were measured by ImageJ software (NIH) at 0, 6, 12, 24 and 48 h.

### 2.12. Cell Invasion and Migration Assay

HSC1 cells, HSC1 EBV cells, SCC25 cells, and SCC25 EBV cells were seeded in the upper chamber of Transwell Chambers (BD Biosciences, Franklin Lakes, NJ, USA) at a density of 5.0 × 10^5^ cells/well in serum-free DMEM in 24-well plates. DMEM containing 20% FBS was applied to the lower chamber as chemoattractant. After 24 h incubation at 5% CO_2_, noninvasive cells on the upper surface of the membrane were removed by wiping with cotton-tipped swabs. Cells that invaded through the matrix gel and attached to the lower surface of the filter were fixed with 10 N Mild-form® for 2 min, permeabilized with methanol for 20 min, and stained with 0.2% crystal violet for 10 min at room temperature. Cells were washed twice with PBS at each step and slides were covered with cover glasses. Invading cells were photographed and counted from 5 different fields.

The cell migration assay was performed according to the aforementioned protocol, except adding the cells into the 0.8 µm Costar® polycarbonate membrane Transwell® insert (Costar, Cambridge, MA, USA).

### 2.13. Apoptosis Assay

Apoptotic cells were quantified by eBioscience^TM^ Annexin V Apoptosis Detection Kit APC (eBioscience). Cells were treated for 24 h with staurosporine at concentrations of 0, 25, 50 and 100 nM. Cells were stained at room temperature for 15 min with APC Annexin V, washed with binding buffer, stained with 7-AAD, and analyzed by flow cytometer. Cells stained by both Annexin V and 7-AAD were considered late apoptotic cells. Cells only positive for Annexin V staining were considered early apoptotic cells.

### 2.14. Statistical Analysis

The GraphPad Prism software (GraphPad Software Inc., San Diego, CA, USA) was used for all data analysis. Mann–Whitney test was used to analyze and to test whether there was a difference between two independent groups, which was expressed as mean ± standard deviation (SD). All experiments were repeated three or five times. A probability (*p*) value < 0.05 was considered statistically significant.

## 3. Results

### 3.1. Detection of EBV-Positive Cells in OSCC Tissues

We analyzed 165 FFPE OSCC tissue specimens. Table 2 shows the characteristics of these OSCC patients: mean age was 70 years and most patients were female (64.8%). According to histological grade, well-differentiated tumors were the most common histological grade (61.8%), followed by moderately/poorly-differentiated tumors (38.2%). The buccal mucosa was the most common anatomical site of OSCC, followed by tongue, lips and other sites, respectively.

Presence of EBV genome was examined by usual PCR method using primers to detect EBNA1 and LMP1 genes. The prevalence of EBV in OSCC was 41.2% (68 out of 165, Figure 1A). To further confirm the infection of EBV in OSCC, EBER-ISH was performed. EBER-positive signal was predominantly found in the nuclei of tumor infiltrating lymphocytes (Figure 1B middle). However, strong EBER-positive signals were detected in epithelial cells of OSCC tissues (Figure 1B right). This result suggests that EBV contributes to the progress of OSCC.

### 3.2. Establishment of EBV-Infected SCC Cells

To investigate the possibility of EBV infection of OSCC cells, the expression of the high affinity CD21 EBV receptor was examined in human tongue-derived SCC25 cells and human skin-derived HSC1 cells. Although HSC1 cells and SCC25 cells did not express CD21, these cells were infected with EBV after co-cultivation with Akata EBV (Figure 2A, Appendix A). Though efficiency of EBV infection was less than 1 in 10^5^ squamous cells, we have established several G418 resistant clones in both HSC1 cells and SCC25 cells. The representative 2 clones were used for further analysis, which were HSC1 EBV clone 1 (C1) and clone 6 (C6) cells and SCC25 EBV clone 8 (C8) and clone 12 (C12) cells. However, the copy number of EBV genome in HSC1 EBV C1 cells, HSC1 EBV C6 cells, SCC25 EBV C8 cells, and SCC25 EBV C12 cells was 2, 2, 5, and 6, respectively (Figure 2B).

To confirm EBV latency status in OSCC cell lines, Western blotting and RT-PCR were perfomed. HSC1 EBV C1 cells and SCC25 EBV C12 cells expressed EBNA1, EBER1, and EBER2 (Figure 2C,D). LMP1 was not detected in SCC25 EBV C12 cells, but was detected in HSC1 EBV C1 cells (Figure 2C). LMP2A product was not detected in either HSC1 EBV cells or SCC25 EBV cells by Western blotting (Figure 2C), but its transcript was detected by RT-PCR. (Figure 2E). These results indicated that squamous epithelial cells showed latency I or II pattern, similar to the latent infections in EBVaGC and in NPC, respectively.

### 3.3. Induction of Lytic EBV Replication in Latently Infected SCC Cells

The expression of EBV lytic genes, including immediate-early genes (BZLF1 and BRLF1), early gene (BMRF1), and late genes (BALF4 and BLLF1), was examined by qRT-PCR. After TPA/SB treatment, all EBV genes expressed at lytic infection (BZLF1, BRLF1, BMRF1, BALF4, and BLLF1) were up-regulated in HSC1 EBV cells. However, the expression of EBV lytic genes was lower in SCC25 EBV cells than in HSC1 EBV cells (Figure 3A–C). Consistent with the mRNA expression profiles, Zta (product of BZLF1), BMRF1 and gp110 (product of BLLF1) proteins were spontaneously expressed in HSC1 EBV cells, but not in SCC25 EBV cells (Figure 3F upper panel). On the contrary, the SCC25 EBV cells expressing Zta and BMRF1 were few (Figure 3F lower panel). Furthermore, TPA/SB treatment significantly increased the number of gp350/220-expressing cells in the HSC1 EBV culture, but not in the SCC25 EBV one (Figure 3D).

Infectious viral titers were calculated by detecting EBV-eGFP infected Daudi cells with flow cytometry. The EBV titer of HSC1 EBV C1 cells was 1.76 × 10^6^ particles/ml (Figure 3E). On the other hand, eGFP-positive Daudi cells were not detected in Daudi (-) incubation with culture supernatant from SCC25 EBV cells. These results demonstrated that the infection of EBV in HSC1 cells and SCC25 cells was productive and abortive, respectively.

### 3.4. EBV Infection Promotes Cell Proliferation and Inhibits Programmed-Cell Death in SCC Cells.

It is reported that EBV infection inhibited apoptosis in epithelial malignancies, such as NPC and EBVaGC [3,16]. We showed that EBV infection significantly induced cell proliferation in HSC1 EBV cells, but not in SCC25 EBV cells (Figure 4). Furthermore, when HSC1 cells and SCC25 cells were treated with staurosporine, the number of apoptotic cells were less in EBV-infected cells than in EBV-uninfected parental cells (Figure 5, Appendix A). These results showed that EBV infection inhibited apoptosis in SCC cells.

### 3.5. EBV Induces Mesenchymal Properties of OSCC Cells

EMT (epithelial-mesenchymal transition) is the transcriptional reprogramming of epithelial cells that contribute to cancer progression and metastasis characterized by inhibition of adhesion and activation of migration and invasion [25]. It is reported that EBV infection induces EMT in NPC cells [3].

EBV infection significantly down-regulated the expression of epithelial markers such as E-cadherin (CDH1) and ZO-1 in both HSC1 EBV cells and SCC25 EBV cells (Figure 6A). Moreover, mesenchemal markers such as ZEB1 and SNAIL1 were significantly up-regulated in both HSC1 EBV cells and SCC25 EBV cells (Figure 6B). EBV infection also upregulated the expression of other mesenchymal markers such as N-cadherin (CDH2) and Vimentin (VIM) in SCC25 EBV cells; however, these genes were down-regulated in HSC1 EBV cells (Figure 6B). These results indicated that EBV infection induced EMT in SCC cells.

### 3.6. EBV Promotes Migration and Invasion in SCC Cells.

Cell migration was examined by wound healing assay and transwell assay. EBV infection significantly promoted migration in both HSC1 EBV C1 cells and SCC25 EBV C12 cells as early as 12 h after incubation (Figure 7A,B). Similarly, EBV infection induced cell invasion in both HSC1 EBV cells and SCC25 EBV cells (Figure 7C). These results showed that EBV promotes tumor metastasis by inducing migration and invasion in SCC cells.

## 4. Discussion

Since EBV is orally transmitted via saliva, many types of oral cells must be infected with EBV. There are many reports that EBV is involved in the development of oropharyngeal cancer. An immortalized keratinocyte-based model has been shown to be susceptible to EBV [26,27]. Here we show for the first time that EBV establishes a latent infection in two squamous epithelial cells, which were well differentiated HSC1 cells and poorly differentiated SCC25 cells.

EBV infection induced cell proliferation, EMT, migration, invasion, and inhibition of apoptosis in both SCC and HSC1 cells. However, EBV-infection promoted proliferation in HSC1 cells, but not in SCC25 cells. The degree of lytic EBV induction also varied from cell to cell. We speculated that the transmission of infection-associated signal was different depending on the status of cell differentiation. In our preliminary microarray analysis, EBV infection up-regulated expression of genes in metabolic pathways, cancer pathways, MAPK pathway, and PI3k/Akt pathway (not shown). In the following paragraphs, we will discuss differences between HSC1 EBV cells and SCC25 EBV cells with regard to latent to lytic EBV activation, apoptotic resistance, cell proliferation, and dedifferentiation (EMT).

Complete EBV activation was observed in well-differentiated HSC1 cells, but not in poorly differentiated SCC25 cells. The observation was well correlated with the report that lytic EBV activation only occurs in differentiated cells, but not in undifferentiated ones [28]. Lytic EBV activation is induced by KLF4, a transcription factor that binds to Zta promoter and is up-regulated in differentiated cells [29]. We have confirmed up-regulation of KLF4 in EBV infected HSC1 cells (Appendix A). In addition to KLF4, activation of other transcription factors that induce Zta was reported in EBV-infected epithelial cells [30,31,32]. As shown in Figure 3, the status of EBV infection in HSC1 cells and SCC25 cells was productive and abortive, respectively. KLF4 seems to be quite important for activation of lytic EBV replication in SCC cells, because the degree of KLF4 expression was well correlated with the degree of EBV gene expression (Figure 3).

EBV products inhibit apoptosis of infected cells in various manners. LMP1 conferred apoptotic resistance by inhibiting the DNA damage response and pro-apoptotic protein level synthesis by activating cellular pathways, especially the PI3K/Akt pathway [33,34,35]. In addition to LMP1, EBV-miRNAs also inhibit apoptosis of cancer cells [36,37]. EBV-miR-BART4 and EBV-miR-BART5-3p inhibited apoptosis by targeting genes, such as PTEN and p53 [38,39]. EBV-infection confers resistance to apoptosis to both HSC1 cells and SCC25 cells (Figure 5). Because SCC25 EBV cells do not express LMP1 protein (Figure 2C), the inhibition of apoptosis by EBV is probably mediated by both LMP1 and EBV-miRNAs.

EBV infection promoted cell proliferation of HSC1 cells, but not of SCC25 cells (Figure 4). We speculated that the balance of EBV gene products could affect cell proliferation, because the expression pattern of EBV gene was different between the two cells. Both EBNA1 and LMP1 was expressed in HSC1 cells, but only EBNA1 was expressed in SCC25 cells (Figure 2). It is reported that expression of EBNA1 promoted tumor formation in the mouse model [40]. On the other hand, LMP1 stimulated cell proliferation by various mechanisms, including up-regulation of PI3K/Akt pathway [41,42]. Promotion of cell proliferation by EBV infection was only observed in HSC1 EBV cells, but not in SCC25 EBV cells. Therefore, LMP1 might stimulate proliferation of EBV infected SCC cells.

EMT contributes to cancer progression and metastasis by inducing migration, invasion, and inhibition of cell adhesion [27]. In the present study, EBV infection induced mesenchymal phenotypes, migration, and invasion of infected cells (Figure 6 and Figure 7). In other EBV-associated epithelial cancers, the activation of EMT was mediated by transcriptional regulators of EMT, such as ZEB1, SNAIL, and TWIST, which was followed by the induction of mesenchymal markers and the suppression of epithelial markers through the activation of cell signaling pathways, such as NF-κB, mTOR, Syk, and PI3K/Akt pathways [43,44,45,46,47,48,49,50,51]. Activation of transcriptional regulators of EMT was followed by down-regulation of epithelial markers and up-regulation of mesenchymal markers in SCC25 EBV cells. However, HSC1 EBV cells showed a distinctive phenotype. We could not observe up-regulation of mesenchymal markers in HSC1 EBV cells. However, the expression of transcriptional regulators of EMT, such as SNAIL1, ZEB1, and integrin-αV, was dramatically increased in HSC1 EBV cells. We are assuming that the status of cell differentiation might affect phenotypic difference, which is becoming an important issue that needs to be investigated in the near future.

Taken together, the present results showed that EBV infection causes neoplastic changes in squamous epithelial cells, and future studies on the molecular mechanisms of tumorigenesis will help develop therapeutic and prophylactic methods for EBV-related oral squamous cell carcinoma.

## 5. Conclusions

This is the first report that mentioned successful EBV infection in SCC cells with a low copy number of viral genomes. Cell differentiation stage was speculated to play a significant role in the regulation of EBV life cycle, and our results showed that EBV infection changes gene expression patterns of SCC cells. The alteration of gene expression may promote the progression of SCC by inducing mesenchymal properties, migration and invasion of cancer cells. In addition, EBV infection inhibits apoptosis of cells. Our results should help understand the relationship between EBV infection in SCC and the oncogenic process of OSCC. However, the molecular mechanisms underlying carcinogenesis by EBV infection still have many unsolved questions. Therefore, further investigations are necessary to clarify the mechanisms by which EBV promotes OSCC.

## Figures and Tables

**Figure 1 microorganisms-08-00419-f001:**
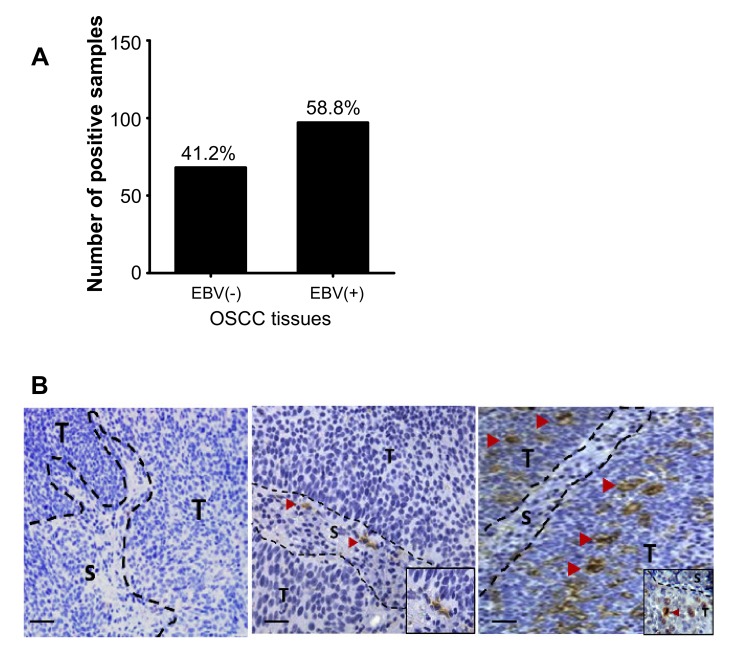
Epstein–Barr virus (EBV) infection detected in OSCC tissues. The presence of EBNA1 and/or LMP1 DNA in OSCC tissues was examined by real-time PCR (**A**). EBNA1 and/or LMP1 positive cases were subjected to further staining for EBER using EBER-ISH. EBER staining divided OSCC into three groups: EBER-negative OSCC (**B**). EBER-staining in OSCC tissues. No EBER-signals either in stromal and tumoral regions (left), EBER-positive infiltrating lymphocytes (Red arrowhead, middle), EBER-positive tumor cells (Red arrowhead, right). EBER positive signals are stained as dark brown in the nuclei and indicated by arrowheads. S: stromal cells, T: tumor cells. Scale bar: 200 μm.

**Figure 2 microorganisms-08-00419-f002:**
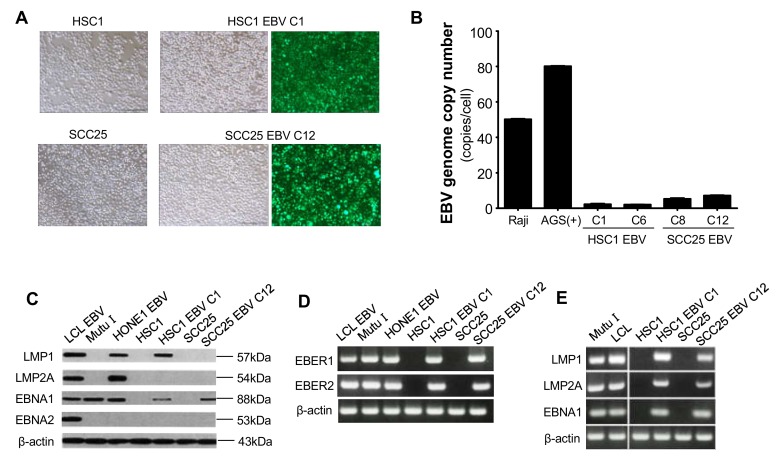
Characterization of EBV-infection in OSCC cells. Establishment of EBV-positive SCC cell lines by cell mediated infection (**A**). Quantification of EBV genome copy number by qPCR (**B**). EBV latency pattern examined by Western blotting (**C**). Expression of EBER1 and EBER2 by RT-PCR (**D**). Expression of LMP1, LMP2A, and EBNA1 by RT-PCR (**E**). Scale bar: 200 μm.

**Figure 3 microorganisms-08-00419-f003:**
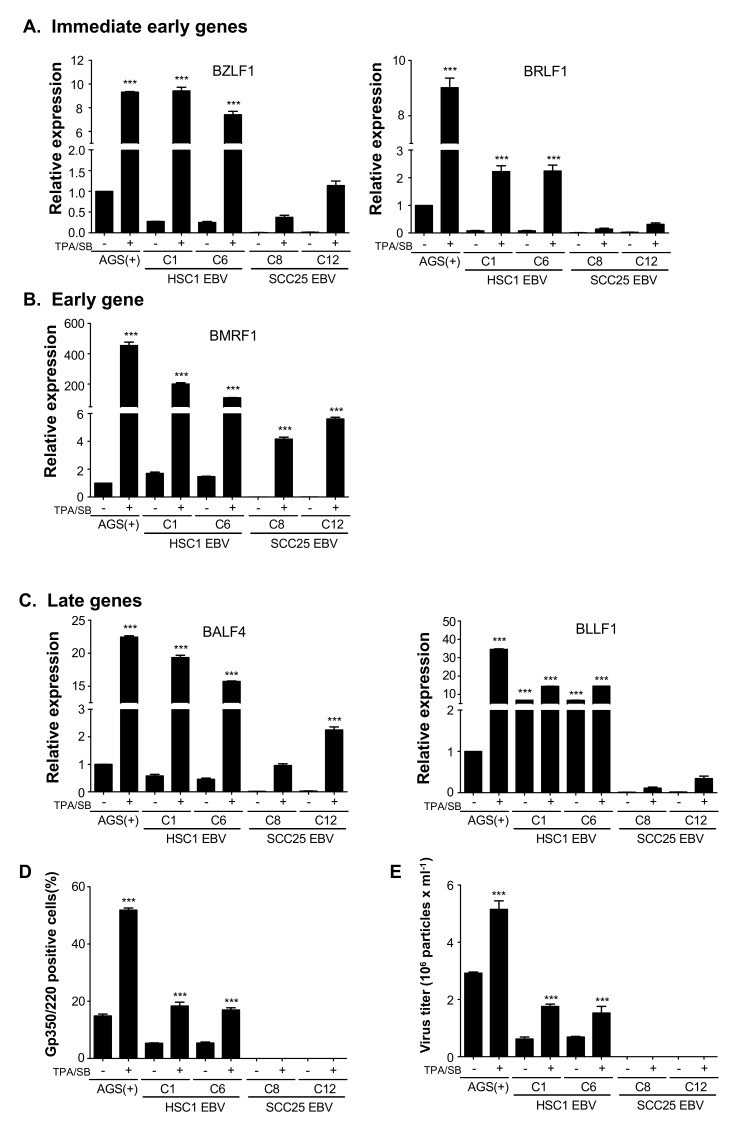
EBV spontaneously activates lytic replication in the differentiated cells. The expression levels of lytic genes, including immediate early genes (**A**) BZLF1, BRLF1; early genes (**B**) BMRF1, and late genes (**C**) BALF4, BLLF1 were analyzed by qRT-PCR. EBV glycoprotein (gp350/220) positive cells were quantified by flow cytometry (**D**). The amount of virus produced from individual cells was calculated by flow cytometry (**E**). The expression of EBV lytic proteins, including Zta, BMRF1 and gp110, was examined by immunofluorescent staining (**F**). ***: *p* < 0.001. Scale bar: 30 μm.

**Figure 4 microorganisms-08-00419-f004:**
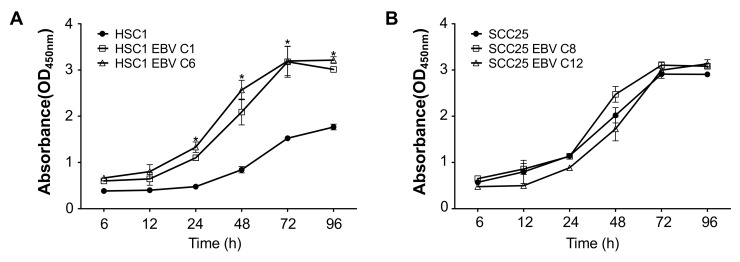
EBV infection induces cell proliferation in HSC1 cells, but not in SCC25 cells. Proliferation of HSC1 cells (**A**) and SCC25 cells (**B**) was examined by CCK-8. *: *p* < 0.05.

**Figure 5 microorganisms-08-00419-f005:**
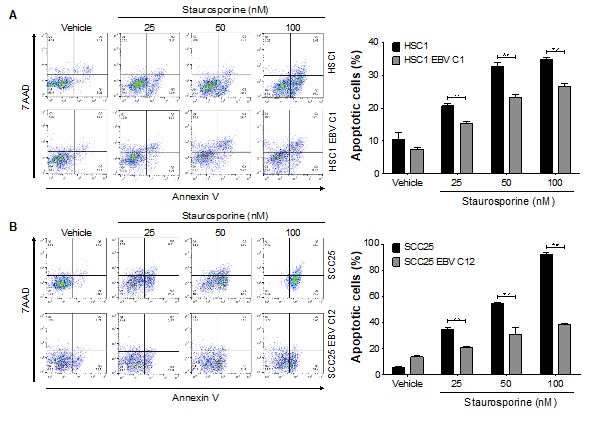
An anti-apoptotic effect of EBV in SCC cells. Both EBV-positive and EBV-negative HSC1 cells (**A**) and SCC25 cells (**B**) were treated with staurosporine at 0, 25, 50 and 100 nM for 24 h. Apoptotic cells were quantified by flow cytometery. **: *p* < 0.01.

**Figure 6 microorganisms-08-00419-f006:**
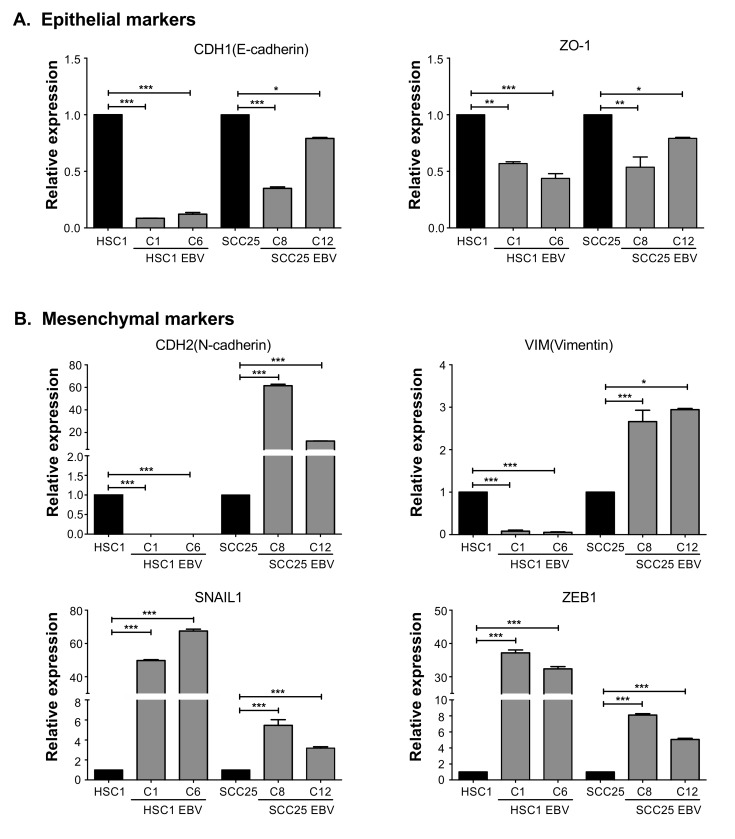
EBV infection induces mesenchymal properties in SCC cells. The expression of genes associated with EMT, including epithelial markers (**A**) CDH1, ZO-1, and mesenchymal markers (**B**) CDH2, VIM, ZEB1, and SNAIL1, was quantitated by qRT-PCR. *: *p* < 0.05; **: *p* < 0.01; ***: *p* < 0.001.

**Figure 7 microorganisms-08-00419-f007:**
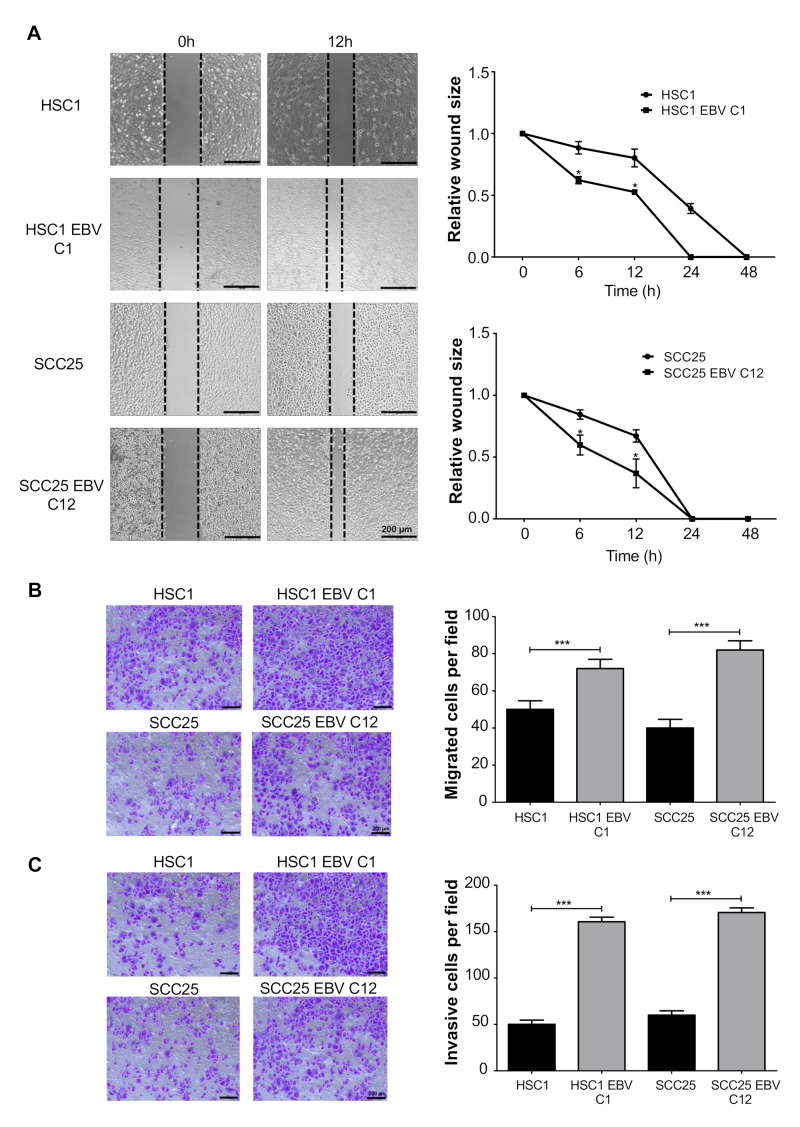
EBV infection induces cell migration and invasion. The effect of EBV infection on cell migration in EBV-infected HSC1 cells and SCC25 cells was examined by wound healing assay (**A**) and transwell assay (**B**). Invasion of EBV-infected HSC1 cells and SCC25 cells was analyzed by transwell assay (**C**). *: *p* < 0.05; **: *p* < 0.01; ***: *p* < 0.001. Scale bar: 200 μm.

**Table 1 microorganisms-08-00419-t001:** Primer sequences.

Gene Name	Forward (5′- -3′)	Reverse (5′- -3′)
qRT-PCR Primers		
*BRLF1*	GGCCCAAAAATTGCAGATGT	CCCACGGGCGAGAATG
*BZLF1*	TCCGACTGGGTCGTGGTT	GCTGCATAAGCTTGATAAGCATTC
*BMRF1*	GCCGTTGAGGCCCACGTTGT	TGGGAATGGCAGGCGAGGGT
*BALF4*	AACCTTTGACTCGACCATCG	ACCTGCTCTTCGATGCACTT
*BLLF1*	GCCGTTGAGGCCCACGTTGT	TGGGAATGGCAGGCGAGGGT
*EBER1*	CTACGCTGCCCTAGAGGTTTT	CAGCTGGTACTTGACCGAAGA
*EBER2*	CCTAGTGGTTTCGGACACACC	AATAGCGGACAAGCCGAATAC
*LMP1*	TCCAGAATTGACGGAAGAGGTT	GCCACCGTCTGTCATCGAA
*LMP2A*	CTCCCTACTCTCCACGGGAT	AGGTAGGGCGCAACAATTACA
*EBNA1*	GGTCGTGGACGTGGAGAAAA	GGTGGAGACCCGGATGATG
*ZO-1*	GAAGGAGTTGAGCAGGAAATCTA	AGGACTCAGCAGTGTTTCACC
*CDH1*	AAGTGCTGCAGCCAAAGACAGA	AAATTGCCAGGCTCAATGACAAG
*CDH2*	TCAGTGGCGGAGATCCTACT	GTGCTGAATTCCCTTGGCTA
*VIM*	CGCCATCAACACCGAGTTC	ATCTTATTCTGCTGCTCCAGGAA
*SNAIL1*	CAGGACTCTAATCCAGAGTTTACCT	ACAGAGTCCCAGATGAGCATTG
*ZEB1*	CATCTTGAGCTGAATTTGGGTAACA	CCTGAAATGACCTGAAGCATGAA
*KLF4*	CATCTTTCTCCACGTTCGCGT	CGGATCGGATAGGTGAAGCTG
*GAPDH*	AATCCCATCACCATCTTCCA	TGGACTCCACGACGTACTCA
β-actin	TGCCGACAGGATGCAGAA	GCCGATCCACACGGAGTACT
**qPCR Primers**		
*EBNA1 gene*	GGAGCCTGACCTGTGATCGT	TAGGCCATTTCCAGGTCCTGTA
*LMP1 gene*	TCTCCTTTGGCTCCTCCTGT	TCGGTAGCTTGTTGAGGGTG
*EBV genome*	ATGTAAATAAAACCGTGACAGCTCAT	TTACCCAACGGGAAGCATATG
β-actin *gene*	GCCATGGTTGTGCCATTACA	GGCCAGGTTCTCTTTTTATTTCTG

**Table 2 microorganisms-08-00419-t002:** Demographic data of oral squamous cell carcinoma (OSCC) patients who provided archived samples.

Characteristic	FFPE OSCC Tissues N = 165
Gender (%)	
Female	107 (64.8)
Male	58 (35.2)
Median age of patients (range)	70 (20–91)
Histology grade of patients (%)	
Well differentiated, EBV positive	49 (29.7)
Well differentiated, EBV negative	53 (32.1)
Moderately + poorly differentiated, EBV positive	19 (11.5)
Moderately + poorly differentiated, EBV negative	44 (26.7)
EBV status (%)	
EBV positive	68 (41.2)
EBV negative	97 (58.8)
Location of tumor (%)	
Buccal mucosa	65 (39.4)
Tongue	54 (32.7)
Lip	30 (18.2)
Other	16 (9.7)

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
