# Peer review of "Epstein–Barr Virus Infection of Oral Squamous Cells"

_microorganisms, 2020, doi:10.3390/microorganisms8030419_

Round 1

Reviewer 1 Report

Epstein-Barr virus infection of oral squamous cells by Heawchaiyaphum et al describes successful infection of SCC cells using a cell-to-cell infection method via co-culture with activated Akata cells, a technique described by Imai, Nishikawa and Takada (reference 20).  The authors create 2 latently infected cell lines EBV-HSC1 and EBV-SCC25 and demonstrate expression of latency genes and viral genome copy numbers.  Spontaneous viral reactivation was observed with EBV-HSC1 but not in EBV-SCC25.  EBV infection of HSC1 cells activated proliferation and migration but only migration in EBV infected SCC25. The authors conclude that EBV can establish latent infection in squamous cells, but migration, proliferation, and lytic induction and infectious virus production may be dependent on cell differentiation (HSC1 cells are well differentiated and SCC25 poorly differentiated). In general, the figures are well done, contain the proper controls, and are convincing.

Minor concerns.

English editing is needed, especially in the introduction.  There are many spelling mistakes (see lines 210 “usig”, “neither” should be “either in line 224”, “caliculated” in line 243.  Titer is spelled ”titter” several times.  Line 241 – “treatment was significantly expressed” – what is meant here?

Figure 1B.  3 groups were described: No EBER staining, EBER staining in stromal region, or EBER staining in tumor region. Of the samples that were EBNA1 and/or LMP1 positive how many of these fell into each group?  Was there fairly equal representation or was one group more common than the others?

Figure 2B.  The presence of HSC1 EBV and SCC25 EBV are shown in green.  It is not explained why these cells are green.  I assume they express GFP but the authors do not state if the co-cultured Akata cells contain a GFP recombinant.  This should be described and be included in the figure legend.  It would also be helpful to see the absence of GFP with the HSC1 and SCC25 uninfected cells as well.

Figure 3, 4, 6.  What are C1, C6, C8 and C12?  These should be described.

Author Response

  We have completed a professional editing service to improve our English. We are very sorry for many mistypes. Line 211; “usig” has been corrected to “using”, which is appearing at line 214 of the revised manuscript. Line 216; “neither or” has been corrected to “not either or” at line 232 of the revised manuscript. Line 243; “caliculated” has been corrected to “calculated”, which is appearing at line 252 of the revised manuscript. Line 245 and 246; “titter” has been corrected to “titer”, which is appearing at line 252 and 253 of the revised manuscript. Line 243; “treatment was significantly expressed” was a mistake. We have deleted “was” to make sense at line 250 of the revised manuscript.

Q1.  Figure 1B.  3 groups were described: No EBER staining, EBER staining in stromal region, or EBER staining in tumor region. Of the samples that were EBNA1 and/or LMP1 positive how many of these fell into each group?  Was there fairly equal representation or was one group more common than the others?

A1. To examine the status of EBV in OSCCs, we firstly assessed the presence of EBV DNA in theses samples using PCR-based technique targeting EBNA-1 and LMP-1. Our flamework and results were shown as below:

This schematic representation of EBV DNA status of 165 OSCCs analyzed by PCR. A proportion of cases with relatively high (EBVDNA++) and with low (EBVDNA+) EBV DNA were selected to EBER ISH study (EBVDNA ++: samples from which both EBNA1 and LMP1 genes were successfully amplified by PCR; EBVDNA+: Samples from which EBNA1, but not LMP1 was successfully amplified by PCR). Therefore, the EBNA1 and/or LMP1 positive samples that fell into each group of EBERs staining were as follows:

EBV DNA negative, n(%)

EBV DNA positive, n (%)

Total, n (%)

EBNA-1+/LMP1-

EBNA-1+/LMP1+

No EBERs

42 (52.5)

0 (0)

0 (0)

42 (52.5)

EBERs-TILs*

0 (0)

29 (36.25)

6 (7.5)

35 (43.75)

EBERs-Tumors

0 (0)

1 (1.25)

2 (2.5)

3 (3.75)

Total

42 (52.5)

30 (3.75)

8 (10)

80 (100)

      *TILs: Tumor-infiltrating lymphocytes

Q2. Figure 2B.  The presence of HSC1 EBV and SCC25 EBV are shown in green.  It is not explained why these cells are green.  I assume they express GFP, but the authors do not state if the co-cultured Akata cells contain a GFP recombinant.  This should be described and be included in the figure legend.  It would also be helpful to see the absence of GFP with the HSC1 and SCC25 uninfected cells as well.

A2. We are sorry for not explaining why EBV-infected cells become GFP positive. It is because we used recombinant EBV having eGFP gene at viral BXLF1 gene. Details are explained in the section of Materials and Methods (Line 79-81 in the revised manuscript).

Q3. Figure 3, 4, 6.  What are C1, C6, C8 and C12?  These should be described.

A3. We have used two EBV infected clones for both HSC1 cells (clone 1 and 6) and SCC25 cells (clone 8 and 12). They are abbreviated to C1, C6, C8, and C12, respectively, and explained from line 224 to line 227 in the revised manuscript.

Reviewer 2 Report

In the present manuscript, Heawchaiyaphum and collegues analyzed the EBV infection of oral squamous carcinoma cells (OSCC). In the first part of the paper they analized by ISH and by real time PCR the presence of EBV in oral speciements from OSCC patients. Furthermore they studied the involvement of EBV in the pathogenesis and the progression of OSCCs by creating in vitro model of EBV infection using SCC cells. The paper is well written and results are clear. According to my opinion some review points needed.

1) my curiosity, no need to answer in the paper: did you checked EBV lytic replication in patient speciements? In the past it has been proven that in OHL speciements there are high level of EBV replication (https://doi.org/10.1002/jmv.10561)

2) among EBV positive samples, how many positives are well differentiated and how many are poorly differentiated? This can be added in table 2.

3) lane 28: please correct "infect to squamous" in "infect squamous"

4) lane 139: please specify which anti gp110 antibody was used

5) lane 200: do you mean real time-PCR or nomal PCR?

6) lane 225: please correct indidated  indicated

7) lane 243: please correct viral titter in viral titer and caliculated in calculated

8) lane 244:  please correct titter in titer

9) lane 318: do you mean dedifferentiated or differentiated?

10) lane 365: please correct inducting

11) Fig 3 please choose a similar focus for TPA(-) and TPA(+) cells, and improve the quality of the IF image.

Author Response

Reviewer #2:

  We have improved our English. And our collective responses are as follows:

Q1. My curiosity, no need to answer in the paper: did you checked EBV lytic replication in patient specimens? In the past it has been proven that in OHL specimens there are high level of EBV replication (https://doi.org/10.1002/jmv.10561).

A1. Yes, we did. In the previous publication, it has been demonstrated that high level of lytic EBV gene products were expressed in EBV-related oral diseases (OHL and OSCC). Similar to this observation, we detected BZLF1 gene in the DNA from EBV-positive OSCC samples, moreover, BZLF1 mRNA from EBER-positive samples by qRT-PCR.

Q2. Among EBV positive samples, how many positives are well differentiated and how many are poorly differentiated? This can be added in table 2.

A2. The number of EBV-positive and -negative in well differentiated OSCC and moderately or poorly differentiated OSCC were shown in the revised Table 2.

Q3. Lane 28: please correct "infect to squamous" in "infect squamous".

A3. Thank you for pointing out our mistake. We have corrected.

Q4. Lane 139: please specify which anti gp110 antibody was used.

A4. We have used C30-1, a monoclonal antibody against gp110, which is also used in the previous paper (Ref. 23). Monoclonal antibodies used were described in the section of immunofluorescence staining (Lanes 141 and 142 in the revised manuscript).

Q5. Lane 200: do you mean real time-PCR or normal PCR?

A5. We used normal PCR method. To avoid uncertainty, we have used the words, “usual PCR method”, at lane204. We also modified the primer list (Table1) to distinguish RT-PCR primers from normal PCR primers.

Q6. Lane 225: please correct ‘indidated’ to ‘iindicated’.

A6. Thank you for pointing out our mistake. We have corrected (Lane 233 in the revised manuscript).

Q7. Lane 243: please correct ‘viral titter’ to ‘viral titer’ and’ caliculated’ to ‘calculated’.

A7. Thank you for pointing out our mistakes. We have corrected (Lane 252 in the revised manuscript).

Q8. Lane 244:  please correct ‘titter’ to ‘titer’.

A8. Thank you for pointing out our mistake. We have corrected (Lane 253 in the revised manuscript).

Q9. Lane 318: do you mean ‘dedifferentiated’ or ‘differentiated’?

A9. Thank you for notifying. To avoid misunderstanding, we used the word ‘well-differentiated’ at lane 326 in the revised manuscript.

Q10. Lane 365: please correct ‘inducting’.

A10. Thank you for pointing out our mistake. We have corrected (Lane 374 in the revised manuscript).

Q11. Fig 3, please choose a similar focus for TPA(-) and TPA(+) cells, and improve the quality of the IF image.

A11. We apologize for the poor resolution of the image and using different magnifications. We have improved Fig. 3F to show the difference in EBV activation between TPA(-) culture and TPA(+) culture.
